# Assessment of severity scoring systems for predicting mortality in critically ill patients receiving continuous renal replacement therapy

**Hyunmyung Park[1], Jihyun Yang[2], Byung Chul Chun[3]***

**1** Department of Epidemiology and Health Informatics, Graduate School of Public Health, Korea University, Seoul, Korea, **2** Department of Internal Medicine, Kangbuk Samsung Medical Center, Seoul, Korea, **3** Department of Preventive Medicine, Korea University College of Medicine, Seoul, Korea

* chun@korea.ac.kr

## Abstract

The incidence of acute kidney injury (AKI) is increasing every year and many patients with AKI admitted to the intensive care unit (ICU) require continuous renal replacement therapy (CRRT). This study compared and analyzed severity scoring systems to assess their suitability in predicting mortality in critically ill patients receiving CRRT. Data from 612 patients receiving CRRT in four ICUs of the Korea University Medical Center between January 2016 and November 2018 were retrospectively collected. The mean age of all patients was 67.6 ± 14.8 years, and the proportion of males was 59.6%. The endpoints were in-hospital mortality and 7-day mortality from the day of CRRT initiation to the date of death. The Program to Improve Care in Acute Renal Disease (PICARD), Demirjian's, Acute Physiology and Chronic Health Evaluation (APACHE) II, Simplified Acute Physiology Score (SAPS) 3, Sequential Organ Failure Assessment (SOFA), Multiple Organ Dysfunction Score (MODS), and Liano's scores were used to predict mortality. The in-hospital and 7-day mortality rates in the study population were 72.7% and 45.1%, respectively. The area under the receiver operator characteristic curve (AUROC) revealed the highest discrimination ability for Demirjian's score (0.770), followed by Liano's score (0.728) and APACHE II (0.710). The AUROC curves for the SAPS 3, MODS, and PICARD were 0.671, 0.665, and 0.658, respectively. The AUROC of Demirjian's score was significantly higher than that of the other scores, except for Liano's score. The Hosmer-Lemeshow test on Demirjian's score showed a poor fit in our analysis; however, it was more acceptable than general severity scores. Kidney-specific severity scoring systems showed better performance in predicting mortality in critically ill patients receiving CRRT than general severity scoring systems.

## Introduction

Acute kidney injury (AKI) occurs in 15%–38% of hospitalized patients, with an in-hospital mortality rate of 23.9%–60.3%, especially in critically ill patients, with a high incidence of up to

---

---

board (e-mali: selfmaster@korea.ac.kr) for researchers who meet the criteria for access to confidential data.

**Funding:** The authors received no specific funding for this work.

**Competing interests:** The authors have declared that no competing interests exist.

74.5% [1–6]. Furthermore, AKI causes additional complications and aggravates the underlying disease, leading to increased hospital stay duration and medical costs [1, 2].

Despite continued progress in medical technology, the incidence of AKI is increasing every year [7] and has become a major public health concern [8]. Among the patients with AKI admitted to the intensive care unit (ICU), 72.5% required renal replacement therapy and 80% received continuous renal replacement therapy (CRRT) [6], primarily because this therapy is hemodynamically more stable than intermittent hemodialysis and fluid balance can be easily controlled [9].

However, CRRT has the disadvantages of high cost and requiring skilled personnel due to the risks of arrhythmia, bleeding, and hypotension [9–11]. Therefore, it is necessary to assess patient severity for predicting prognosis and identifying meaningful information necessary for medical staff to discuss and make correct decisions about patient prognosis, and for providing future treatment directions to patients and care givers [12]. Moreover, predicting the mortality rate of patients admitted to the ICU is critical for assessing the severity of the disease and adjudicating the value of new treatments, interventions, and health care policies [13]. Estimates of mortality risk can be useful for the efficient allocation of resources and judgment of treatment adequacy in medical institutions by comparing actual and expected outcomes [14, 15].

The severity scoring system can be divided into assessing the overall health status and measuring severity by focusing on specific organs. Various scoring systems have been developed to predict disease prognosis [16–22]. Several studies have used severity scoring systems to effectively apply CRRT according to the acuity of illness parameters [23, 24]. Additionally, to improve the quality of CRRT, such as the optimal start time of CRRT [25, 26], severity scores are used for providing the objectivity and reliability of the study with population stratification and balanced randomization to ensure that disease severity does not affect the statistical outcome [27]. Although other studies have compared the predictive abilities of severity scoring systems in patients with AKI, most studies included all patients diagnosed with AKI who received renal replacement therapy, such as intermittent hemodialysis, and only few focused solely on CRRT [28–34].

This study aimed to evaluate the predictive ability of severity scoring systems for mortality in critically ill patients receiving CRRT. This study compared and analyzed the Acute Physiology and Chronic Health Evaluation (APACHE) II score, Simplified Acute Physiology Score (SAPS) 3, Sequential Organ Failure Assessment (SOFA) score, and Multiple Organ Dysfunction Score (MODS), which are general severity scoring systems predicting mortality, and Liano's, Program to Improve Care in Acute Renal Disease (PICARD), and Demirjian's scores, which are kidney-specific severity scoring systems. The results of this study could be used as a basis for selecting scoring systems suitable for severity assessment in patients with AKI receiving CRRT, and for developing a new severity scoring system.

## Materials and methods

### Study population

From January 2016 to November 2018, patients who received CRRT in four ICUs at the Korea University of Medicine were selected. The target group included patients with AKI and chronic kidney disease (CKD) who did not undergo dialysis. Patients who had previously undergone hemodialysis or peritoneal dialysis for end-stage renal disease or who had received a kidney transplant were excluded. Of the 768 patients who received CRRT during this period, 612 were included in the final analysis, excluding 119 patients who had previously undergone hemodialysis, 23 who had undergone peritoneal dialysis, and 14 who had kidney transplants.

In cases where CRRT was performed several times during hospitalization, the clinical symptoms and diagnostic test results at the time of initial treatment were investigated.

This study was approved by the Institutional Review Board of the Korea University Anam Hospital (approval number: 2018AN0415). Informed consent was waived by the board as this study was conducted retrospectively and the data were de-identified prior to analysis. All study methods were performed in accordance with relevant guidelines and regulations.

## Data

Clinical data of patients receiving CRRT were retrospectively collected from electronic medical records. The patients' age, sex, in-hospital mortality, survival at 7 days from the initiation of CRRT, comorbidities, reason for CRRT, mean arterial pressure (MAP) at the initiation of treatment, use of vasopressors, mechanical ventilation, and laboratory findings, including serum hemoglobin, serum creatinine, albumin, arterial blood gas analysis, and C-reactive protein levels, were investigated.

The scores of APACHE II, SAPS 3, SOFA, MODS, Liano's, PICARD, and Demirjian's were calculated and used to predict mortality. Severity scoring was performed at the initiation of CRRT, and all variables for severity scores were collected within 24 h prior to the initiation of CRRT. The endpoints were in-hospital and 7-day mortality according to the duration from the day of CRRT initiation to the date of death.

## Statistical analysis

The general characteristics of the participants were recorded using mean value and standard deviation, and the area under the receiver operating characteristic curve (AUROC) of each severity score was calculated to assess discrimination among the severity scoring systems.

Calibration of the severity score was assessed using the Hosmer-Lemeshow Goodness-of-Fit test. Data analysis was performed using IBM SPSS statistical software, version 23 (SPSS Inc., Chicago, IL, USA), and ROC comparisons were performed using the MedCalc statistical software (MedCalc, Ostend, Belgium). The statistical significance level was set at p-value $>0.05$.

## Results

### Baseline characteristics and mortality of the study population

A total of 612 participants were enrolled in the study. The mean age of all participants was $67.6 \pm 14.8$ years and 59.6% were males (Table 1). The in-hospital mortality rate was 72.7% and 7-day mortality was 45.1%. Patients with CKD accounted for 11.9% of the total study population, and there was no significant difference in the proportions of survivors and non-survivors. There were no significant differences between survivors and non-survivors in terms of mean age, sex, and frequency of AKI causes. Table 2 shows the clinical test results and acute physiology of the study population at the time of CRRT initiation. pH, serum albumin, and platelet count in non-survivors were significantly lower than those in survivors ($p < 0.01$ for all).

### Severity scores of study population

The mean severity scores for the study population were as follows: APACHE II score, 35.5; SAPS 3, 84.6; SOFA score, 9.0; MODS, 10.7; Liano's score, 0.55; PICARD score, 0.43; and Demirjian's score, 0.60. Table 3 shows a comparison of the severity scores between survivors and non-survivors. There was a significant difference in the mean of all severity scores between survivors and non-survivors ($p < 0.01$ for all).

**Table 1. Demographics and clinical characteristics of the study population.**

| Parameter | All patients n = 612 (%) | Survivors n = 167 (%) | Non-survivors n = 445 (%) | p-value |
|---|---|---|---|---|
| Age (years) | 67.6 ± 14.8 | 67.4 ± 14.9 | 67.7 ± 14.7 | 0.851* |
| Sex | | | | |
| Male | 365 (59.6) | 95 (56.9) | 270 (60.7) | 0.406+ |
| Female | 247 (40.4) | 72 (43.1) | 175 (39.3) | |
| Etiology of AKI | | | | |
| Sepsis | 400 (65.4) | 108 (64.7) | 292 (65.6) | 0.849+ |
| Nephrotoxic | 226 (36.9) | 54 (32.3) | 172 (38.7) | 0.159+ |
| Ischemic | 204 (33.3) | 52 (31.1) | 152 (34.2) | 0.502+ |
| Others | 62 (10.1) | 18 (10.8) | 44 (9.9) | 0.764+ |
| CKD history | 73 (11.9) | 24 (11.4) | 49 (11.0) | 0.264 |
| Comorbidities | | | | |
| Diabetes mellitus | 231 (37.7) | 81 (48.5) | 150 (33.7) | 0.001+ |
| Hypertension | 336 (54.9) | 99 (59.3) | 237 (53.3) | 0.202+ |
| Heart Failure | 91 (14.9) | 27 (16.2) | 64 (14.4) | 0.610+ |
| Coronary artery disease | 94 (15.4) | 24 (14.4) | 70 (15.7) | 0.708+ |
| COPD | 13 (2.1) | 2 (1.2) | 11 (2.5) | 0.530+ |
| Cancer | 111 (18.1) | 21 (12.6) | 90 (20.2) | 0.034+ |
| Liver disease | 100 (16.3) | 18 (10.8) | 82 (18.4) | 0.027+ |

Continuous data are presented as mean ± SD, and categorial data as number of patients (%). AKI, acute kidney injury; COPD, chronic obstructive pulmonary disease.

*p-value by Student's t-test

+p-value by Fisher's exact test

## Discrimination of each severity scoring system

The AUROCs for in-hospital mortality are shown in Fig 1. AUROC revealed acceptable discrimination ability for Demirjian's score, followed by Liano's score. Table 4 shows the results of the comparison of AUROC between the scoring systems. Demirjian's score was not significantly different from Liano's score but was significantly higher than the rest.

To compare the discrepancies in severity scores according to the survival period, cases of death within 7 days after CRRT initiation were evaluated. The AUROC for 7-day mortality was as follows: APACHE II score, 0.707 (95% CI, 0.666–0.748); SAPS 3, 0.629 (95% CI, 0.585–0.673); SOFA score, 0.590 (95% CI, 0.545–0.635); MODS, 0.651 (95% CI, 0.607–0.694); Liano's score, 0.725 (95% CI, 0.686–0.765); PICARD score, 0.569 (95% CI, 0.523–0.614); and Demirjian's score, 0.768 (95% CI, 0.731–0.805). Similar to in-hospital mortality, Demirjian's score showed a relatively high value in predicting the 7-day mortality.

## Calibration of each severity scoring systems

Fig 2 shows the calibration of each severity score. Two severity scoring systems were excluded: the SOFA score and MODS, which do not generate the probability of death but only count points. Except for Liano's score ($\chi 2$ = 7.555, p = 0.478) and the PICARD score ($\chi 2$ = 14.835, p = 0.062), the Hosmer-Lemeshow test for in-hospital mortality demonstrated a poor fit of the prediction models (p < 0.05). The calibration for 7-day mortality was similar to that for in-hospital mortality. Only SAPS 3 showed different results, and the result was significant for 7-day mortality ($\chi 2$ = 9.224, p = 0.324).

**Table 2. Clinical test results and acute physiology of study population at continuous renal replacement therapy (CRRT) initiation.**

| Parameter | All patients (n = 612) mean ± SD | Survivors (n = 167) mean ± SD | Non-survivors (n = 445) mean ± SD | p-value |
|---|---|---|---|---|
| Serum creatinine (mg/dL) | 3.1 ± 1.9 | 3.8 ± 2.4 | 2.9 ± 1.8 | <0.001** |
| BUN (mg/dL) | 57.8 ± 33.8 | 61.8 ± 37.4 | 56.5 ± 32.6 | 0.236** |
| Urine volume (mL/day) | 759 ± 1003 | 1169.9 ± 1244.9 | 634.0 ± 898.4 | <0.001** |
| Hb (g/dL) | 9.9 ± 2.2 | 10.0 ± 2.1 | 9.9 ± 2.3 | 0.615* |
| WBC count (x10³/μL) | 14.5 ± 9.9 | 15.3 ± 9.2 | 14.4 ± 10.3 | 0.092** |
| Lactic acid (mmol/L) | 7.3 ± 6.1 | 4.9 ± 4.9 | 8.2 ± 6.3 | <0.001** |
| CRP (mg/L) | 125.6 ± 112.3 | 117.7 ± 102.8 | 128.9 ± 115.6 | 0.373** |
| Arterial pH | 7.2 ± 0.14 | 7.4 ± 0.1 | 7.3 ± 0.1 | <0.001** |
| Arterial bicarbonate (mmol/L) | 16.5 ± 5.6 | 17.3 ± 5.9 | 16.2 ± 5.5 | 0.039* |
| Total bilirubin (mg/dL) | 3.4 ± 7.1 | 3.2 ± 8.3 | 3.5 ± 6.5 | <0.001** |
| INR | 1.7 ± 1.1 | 1.5 ± 0.9 | 1.8 ± 1.1 | <0.001** |
| Platelet (x10³/μL) | 110.8 ± 91.8 | 141.4 ± 94.0 | 99.3 ± 88.4 | <0.001** |
| Serum albumin (g/dL) | 2.7 ± 0.6 | 2.9 ± 0.6 | 2.6 ± 0.6 | <0.001* |
| Serum phosphate (mg/dL) | 5.5 ± 2.8 | 5.0 ± 2.5 | 5.7 ± 2.9 | 0.010** |
| MBP (mmHg) | 73.5 ± 16.3 | 79.0 ± 16.1 | 71.4 ± 15.9 | <0.001* |
| Heart rate (bpm) | 110.8 ± 29.3 | 102.8 ± 27.2 | 113.8 ± 29.5 | <0.001* |
| Mechanical ventilation | 395 (64.5) | 71 (42.5) | 324 (72.8) | <0.001+ |
| Vasopressor use | 445 (72.7) | 85 (50.9) | 360 (80.9) | <0.001+ |

Continuous data are presented as mean ± SD, and categorial data as number of patients (%). BUN, blood urea nitrogen; Hb, hemoglobin; WBC, white blood cell; CRP, C-reactive protein; INR, international normalized ratio; MBP, mean blood pressure.

*p-value by Student's t-test

**p-value by Mann-Whitney test

## Discussion

This study evaluated and compared the predictive ability of the severity scores of patients who received CRRT. The primary result was that the kidney severity scores performed better than the general severity scores because of comparing the predictive ability between the severity scores. The AUROC for in-hospital mortality revealed acceptable discrimination ability of the Demirjian's score (0.770), followed by the Liano's (0.728) and APACHE II (0.710) scores. Demirjian's score also showed the highest predictive value for 7-day mortality, followed by the

**Table 3. Comparison of severity score between survivors and non-survivors.**

| Severity score | All patients (n = 612) mean ± SD | Survivors (n = 167) mean ± SD | Non-survivors (n = 445) mean ± SD | p-value |
|---|---|---|---|---|
| APACHE II score | 35.5 ± 9.3 | 30.6 ± 8.8 | 37.3 ± 8.8 | <0.001* |
| SAPS 3 | 84.6 ± 19.9 | 73.6 ± 17.7 | 87.7 ± 19.8 | <0.001* |
| SOFA score | 9.0 ± 3.3 | 8.0 ± 3.3 | 9.4 ± 3.2 | <0.001* |
| MODS | 10.7 ± 4.3 | 9.8 ± 0.5 | 11.1 ± 0.1 | 0.001* |
| Liano's score | 0.55 ± 0.22 | 0.42 ± 0.19 | 0.60 ± 0.21 | <0.001* |
| PICARD score | 0.43 ± 0.16 | 0.37 ± 0.15 | 0.46 ± 0.15 | <0.001* |
| Demirjian's score | 0.60 ± 0.31 | 0.38 ± 0.29 | 0.68 ± 0.28 | <0.001** |

Data are presented as mean ± SD. APACHE, Acute Physiology and Chronic Health Evaluation; SAPS, Simplified Acute Physiology Score; SOFA, Sequential Organ Failure Assessment; MODS, Multiple Organ Dysfunction Score; PICARD, The Program to Improve Care in Acute Renal Disease.

*p-value by Student's t-test

**p-value by Mann-Whitney test

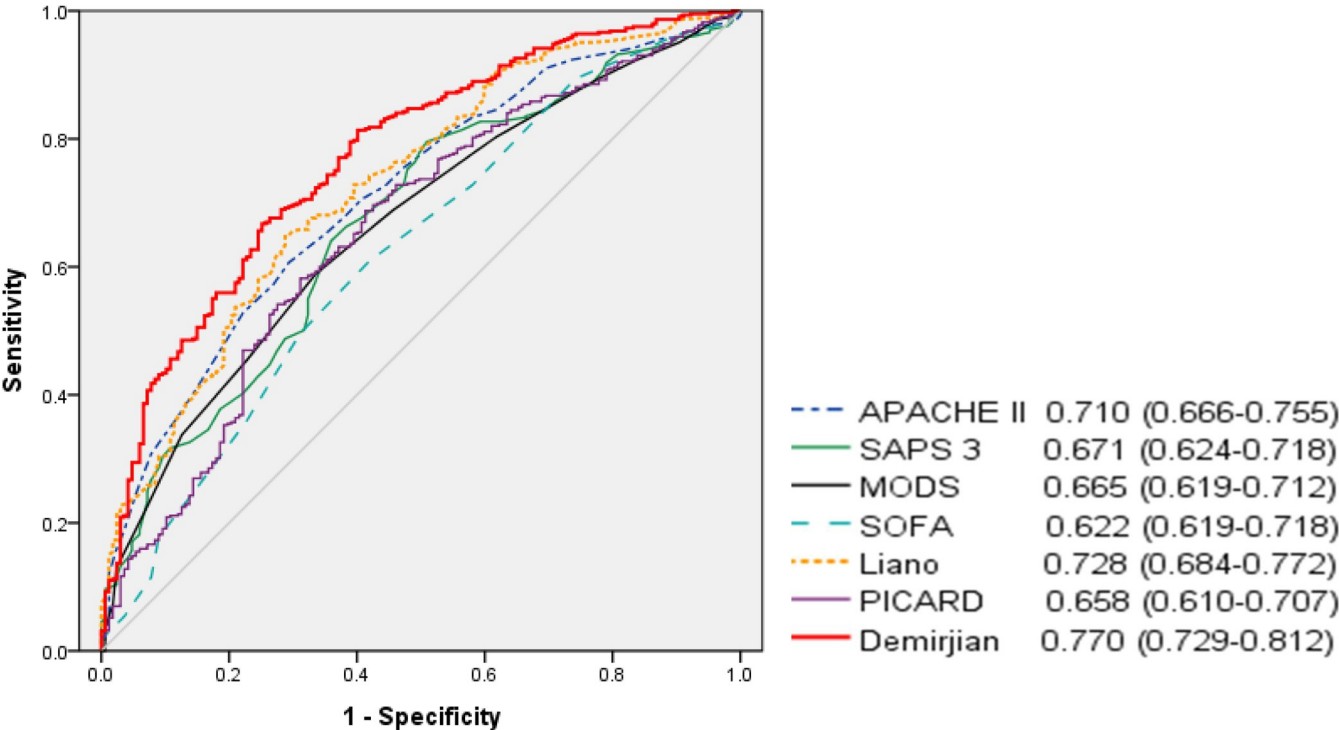

| | APACHE II | 0.710 (0.666-0.755) |
| | SAPS 3 | 0.671 (0.624-0.718) |
| | MODS | 0.665 (0.619-0.712) |
| | SOFA | 0.622 (0.619-0.718) |
| | Liano | 0.728 (0.684-0.772) |
| | PICARD | 0.658 (0.610-0.707) |
| | Demirjian | 0.770 (0.729-0.812) |

**Fig 1. Area under receiver operating characteristic curves (AUROCs) of the seven severity scores for in-hospital mortality.**

Liano's and APACHE II scores. The AUROC comparison showed that Demirjian's score was significantly higher than the other scores except for Liano's score. In addition, the Hosmer-Lemeshow test results of five scores, which provide predicted mortality, showed poor calibration of all scores except for those of Liano's and PICARD. The calibration for 7-day mortality was similar to that for in-hospital mortality.

The in-hospital mortality was 72.7% in this study, which is higher than the 23.9%–60.3% due to AKI [1–6], implying that among patients with AKI, patients undergoing CRRT have a higher mortality rate. In a previous study of 1738 patients with AKI, 76.2% received mechanical ventilation, 69.1% received vasopressors, and 47.5% had sepsis at the onset of CRRT [6]. The use of mechanical ventilation was lower in this study than in a previous study; however, the use of vasopressors and sepsis rates were higher. These differences may be related to differences in mortality rates. In the non-survivor group, the MBP was lower at the start of CRRT, and mechanical ventilation and vasopressor use were more frequent, indicating that vital signs at the beginning of CRRT were worse. However, the mortality rate of our study population, which was relatively higher than that of other studies, is a potential limitation.

The general severity scores evaluated in our study were lower than the AUROC of 0.7, except for the APACHE II score. The APACHE II score was more discriminative than the other general scores; however, the results of the fitness test showed poor calibration, and the calibration line tended to underestimate mortality.

In previous studies that evaluated mortality based on general severity scores in patients with AKI, discriminant assessments were inconsistent. Passos et al. compared the APACHE II score, SAPS 3, and SOFA score in 186 patients with sepsis who underwent CRRT, and the AUROC showed poor discrimination, with 0.57, 0.48, and 0.58, respectively [33]. A study of 1169 patients with AKI in China from 1996 to 2013 showed that the AUROC of the SOFA

**Table 4. Pairwise comparison of receiver operating characteristic curves for in-hospital mortality.**

| Severity score system | | Compared scores | 95% CI | p-value |
|---|---|---|---|---|
| **Demirjian's score** | ~ | Liano's score | -0.005–0.091 | 0.081 |
| | ~ | APACHE II score | 0.009–0.111 | 0.020 |
| | ~ | SAPS 3 | 0.050–0.149 | <0.001 |
| | ~ | MODS | 0.053–0.157 | <0.001 |
| | ~ | PICARD score | 0.065–0.159 | <0.001 |
| | ~ | SOFA score | 0.093–0.204 | <0.001 |
| **Liano's score** | ~ | APACHE II score | -0.029–0.063 | 0.459 |
| | ~ | SAPS 3 | 0.008–0.106 | 0.023 |
| | ~ | MODS | 0.004–0.121 | 0.036 |
| | ~ | PICARD score | 0.006–0.132 | 0.031 |
| | ~ | SOFA score | 0.042–0.017 | 0.001 |
| **APACHE II score** | ~ | SAPS 3 | -0.002–0.082 | 0.064 |
| | ~ | MODS | -0.005–0.095 | 0.076 |
| | ~ | PICARD score | -0.011–0.115 | 0.105 |
| | ~ | SOFA score | 0.028–0.148 | 0.004 |
| **SAPS 3** | ~ | MODS | -0.042–0.054 | 0.816 |
| | ~ | PICARD score | -0.044–0.069 | 0.664 |
| | ~ | SOFA score | -0.005–0.102 | 0.073 |
| **MODS** | ~ | PICARD score | -0.053–0.067 | 0.825 |
| | ~ | SOFA score | 0.001–0.086 | 0.048 |
| **PICARD score** | ~ | SOFA score | -0.025–0.097 | 0.241 |

APACHE, Acute Physiology and Chronic Health Evaluation; SAPS, Simplified Acute Physiology Score; SOFA, Sequential Organ Failure Assessment; MODS, Multiple Organ Dysfunction Score; PICARD, The Program to Improve Care in Acute Renal Disease.

score was 0.78 [34]. Of the 731 patients, only 56.1% underwent RRT, and the overall mortality rate was 13.8%, indicating that the severity of the disease was low.

Liano's score was developed by Liano et al. in Spain [20]. The discriminant ability of Liano's score was higher than that of the other scores, except for Demirjian's score in this study. Additionally, the Hosmer-Lemeshow test on Liano's score showed good calibration. Liano's scores have been evaluated for external validity in several previous studies. Uchino et al. conducted a prospective multinational multicenter study of patients with AKI involving 54 medical institutions in 23 countries from 2000 to 2001 [28]. A total of four kidney-specific severity scores (Mehta [35], Liano's, Chertow [36], and Paganini [37]) and two general severity scores (SAPS II and SOFA) were calculated to compare the predictive ability. The AUROC of Liano's score was 0.698, which was more discriminative than the other scores; however, all were less than 0.7. Calibration was poor for all except for Liano's score in this study.

Maccariello et al. compared the mortality predictive ability of the APACHE II score, SAPS II, Logistic Organ Dysfunction [38], Organ Dysfunction and Infection [39], Liano's score, and Mehta score in 467 patients with AKI who received RRT in ICUs [29]. The AUROC score was above 0.7 for the SAPS II and Mehta scores, and all the scores except the Mehta score showed good calibration. In this study, the high proportion of patients with sepsis (76%) and mechanical ventilation dependence (81%) may have influenced the results. Ohnuma et al., performed a retrospective data analysis of 343 patients with AKI who underwent CRRT in Japan [31]. The mortality predictive external validity of the Mehta, SHARF II [40], PICARD, VELLORE [41], Liano's, and Demirjian's scores, which are kidney-specific severity scores, were compared with

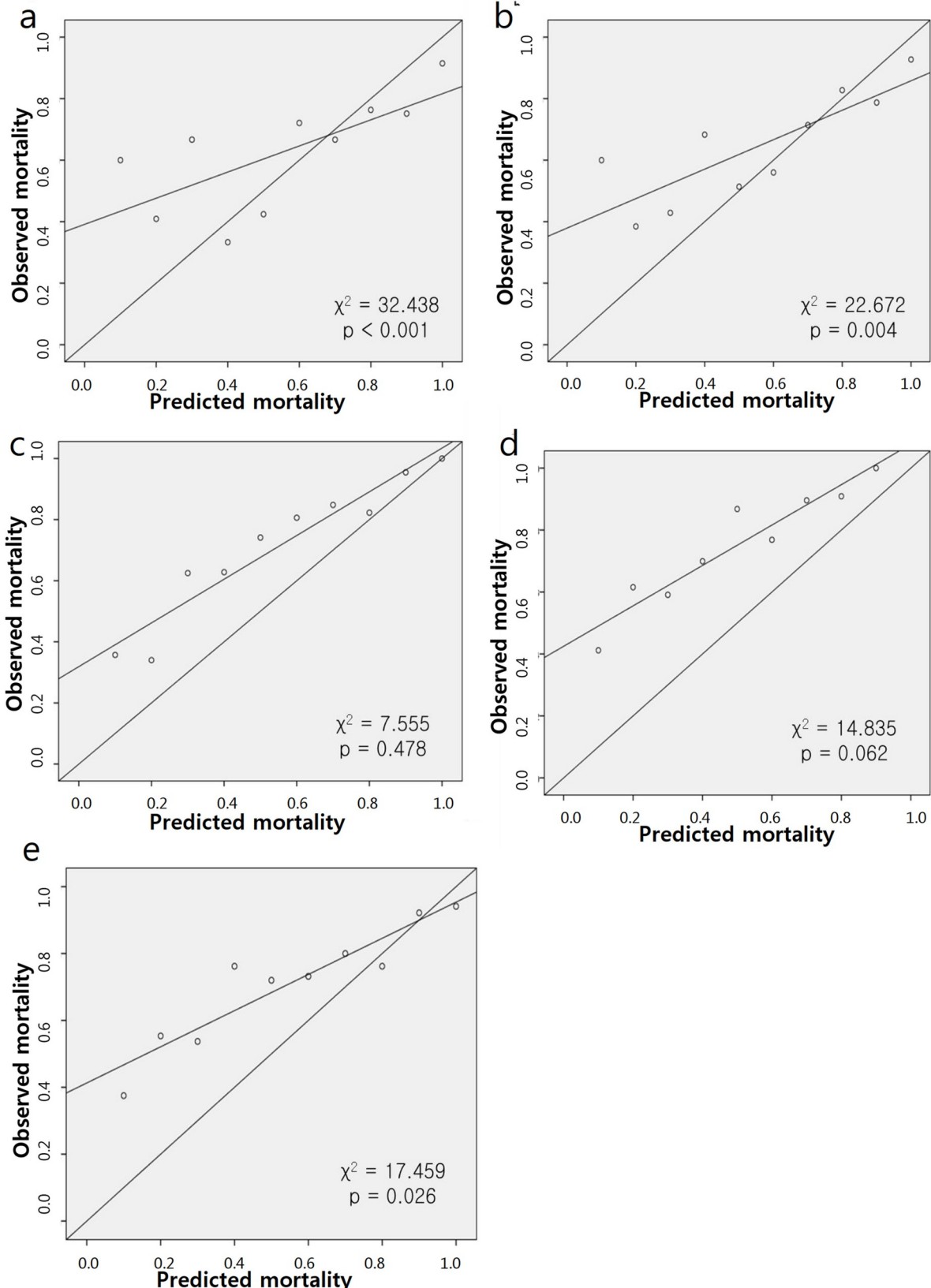

**Fig 2.** Calibration lines of five severity scores for in-hospital mortality: APACHE II (a), SAPS 3 (b), Liano's (c), PICARD (d), and Demirjian's (e).

the SOFA score; all were less than an AUROC of 0.7. The results of the goodness-of-fit tests were poor, except for Liano's score.

The PICARD score was developed by Chertow et al. based on the Program to Improve Care in Acute Renal Disease (PICARD), a multicenter study of 618 patients with AKI in five U.S. medical institutions from 1999 to 2001 [21]. A formula for predicting 60-day mortality was developed by dividing the time of AKI diagnosis, consultation, and initiation of dialysis. In this study, a prediction formula was applied and analyzed based on the dialysis initiation time, where the AUROC was the highest in internal validation. The PICARD score showed good calibration but the lowest discrimination among the kidney-specific severity scores in our analysis. Discrimination is affected by the distribution of the target group, which is poor in the homogeneous group and good in the heterogeneous group [39]. The predicted mortality rate of PICARD was 43%, indicating its tendency to underestimate the mortality rate. This is thought to be attributed to a 60-day mortality criterion and a low mortality rate of 37% among the populations that developed these scores.

Demirjian's score exhibited the highest discriminative ability. Demirjian's score was developed from the Veterans Affairs/National Institutes of Health Acute Renal Failure trial network study in the United States [22] to predict the 60-day mortality by selecting 21 variables affecting mortality among patients with AKI who received CRRT. The Hosmer-Lemeshow test on Demirjian's score showed a poor fit in our analysis; however, it was more acceptable than general severity scores.

Although this study is limited in that it analyzed retrospectively collected data in a single-institution ICU, it has the strength of assessing the mortality predictability of kidney-specific severity scores only in patients who received CRRT. In several previous studies, general severity scores were used for population stratification and balanced randomization to improve the quality of CRRT. For example, Zarbock et al. compared the effect of early and delayed RRT initiation on mortality in critically ill patients with AKI, in which randomization was stratified according to SOFA cardiovascular scores [25]. In the study by Barbar et al. on the timing of RRT in patients with AKI, randomization was performed based on a minimization technique with stratification according to center, age, SOFA score, and type of infection [26]. This study supports the fact that the kidney-specific severity scores have higher discriminative ability than systemic scores in predicting mortality in patients receiving CRRT, and highlights the need to develop more predictable tools for patients with AKI receiving CRRT.

Patients with CKD were included in the study population, except those who received renal replacement therapy, such as intermittent hemodialysis, peritoneal dialysis, and kidney transplantation. Since the focus was on patients receiving CRRT, the study results are unlikely to change due to the characteristics of this cohort; however, the lack of information, such as baseline creatinine or eGFR, is a limitation of this study.

## Conclusions

In summary, compared with general severity scores, kidney-specific severity scores demonstrated better calibration and discrimination in predicting mortality in patients with AKI receiving CRRT. However, none of the parameters evaluated in this study exhibited both excellent differentiation and calibration. In conclusion, all severity scoring systems included in this study had a limited ability to predict mortality in critically ill patients requiring CRRT.

Therefore, we emphasize the need to develop novel severity scores with good calibration and high discrimination abilities.

## Author Contributions

**Conceptualization:** Hyunmyung Park, Jihyun Yang, Byung Chul Chun.

**Data curation:** Hyunmyung Park, Jihyun Yang.

**Formal analysis:** Hyunmyung Park.

**Investigation:** Hyunmyung Park, Jihyun Yang.

**Methodology:** Hyunmyung Park, Byung Chul Chun.

**Project administration:** Byung Chul Chun.

**Resources:** Jihyun Yang, Byung Chul Chun.

**Software:** Hyunmyung Park.

**Supervision:** Byung Chul Chun.

**Validation:** Jihyun Yang, Byung Chul Chun.

**Visualization:** Hyunmyung Park.

**Writing – original draft:** Hyunmyung Park.

**Writing – review & editing:** Hyunmyung Park, Jihyun Yang, Byung Chul Chun.

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
