## [Decision Letter · Decision Letter 0]

9 Feb 2023

PONE-D-22-26854Assessment of severity scoring systems for predicting mortality in critically ill patients receiving continuous renal replacement therapyPLOS ONE

Dear Dr. Chun,

Thank you for submitting your manuscript to PLOS ONE. After careful consideration, we feel that it has merit but does not fully meet PLOS ONE’s publication criteria as it currently stands. Therefore, we invite you to submit a revised version of the manuscript that addresses the points raised during the review process.

**The manuscript focuses as a topic of potential interest. The study, however, has major shortcomings that preclude sound conclusions, and should be addressed. To mention some of them, i) concern about the fact that many of the scores being compared have used in their developmental cohort patients only with AKI needing CRRT unlike author study here where CKD patients needed dialysis is also included; ii) need to mention what percent of their patients had CKD among survivors vs non-survivors; iii) there is no mention of timing of scoring with respect to CRRT start; iv) concern about the fact that the authors overstate how good any of the models are; v) need to further explain that none of these tests are particularly good, and therefore they should more strongly emphasize that the primary conclusion of this study is that additional research is needed to develop better scoring systems; vi) unclear why the mortality of the cohort was so high; vii) need to include this high mortality as a potential limitation of the study, because these results may not be as relevant to centers with different patient selection and/or better CRRT outcomes, viii) need to report the results of the Hosmer-Lemeshow tests by simply providing the terms “significant” or “not significant”; ix) unclear if this is a secondary analysis of a cohort which was generated for a prior study; x) concern about Table 3, which is somewhat confusing; xi) need to review some statements in the Introduction that are somewhat misleading; xii) need to provide the distribution of baseline creatinine or eGFR and the proportion of patients with advanced CKD (stage 4 or 5); xiii) need to provide the relative number of patients with de novo AKI, AKI on CKD, and progressive CKD.**

We look forward to receiving your revised manuscript.

Kind regards,

Giuseppe Remuzzi

Academic Editor

PLOS ONE

Journal Requirements:

2. Please ensure that you have specified (1) whether consent was informed and (2) what type you obtained (for instance, written or verbal, and if verbal, how it was documented and witnessed). If your study included minors, state whether you obtained consent from parents or guardians. If the need for consent was waived by the ethics committee, please include this information.

If your study used retrospective hospital data- in ethics statement in the manuscript please ensure that you have discussed whether all data/samples were fully anonymized before you accessed them and/or whether the IRB or ethics committee waived the requirement for informed consent. If patients provided informed written consent to have data/samples from their medical records used in research, please include this information.

Reviewers' comments:

Reviewer's Responses to Questions

**Comments to the Author**

1. Is the manuscript technically sound, and do the data support the conclusions?

Reviewer #1: Partly

Reviewer #2: Yes

2. Has the statistical analysis been performed appropriately and rigorously? 

Reviewer #1: Yes

Reviewer #2: Yes

3. Have the authors made all data underlying the findings in their manuscript fully available?

Reviewer #1: Yes

Reviewer #2: Yes

4. Is the manuscript presented in an intelligible fashion and written in standard English?

Reviewer #1: No

Reviewer #2: Yes

5. Review Comments to the Author

Reviewer #1: The study by Chun et al is an original research to attempt using well validated scoring systems to predict 7 day and in hospital mortality in patients on CRRT. The study does follow the necessary statistical requirement to compare scoring systems and uses the standard metrics to judge prognostic scoring systems including discrimination and calibration. Unfortunately the conclusions are not valid since there are methodological flaws. Firstly many of the scores being compared have used in their developmental cohort patients only with AKI needing CRRT unlike authors study where CKD patients needing dialysis is also included. Additionally, they do not mention what percentage of their patients had CKD among survivors vs non-survivors. Besides this there is no mention of timing of scoring with respect to CRRT start.

Minor issues: Sentences need to structured for more clarity: for instance

Line 71 : which study is mentioned here

Line 72: Another several studies is grammatically incorrect. I think authors want to say that other studies have compared predictive power of disease severity scoring systems

Line 74 : needs clarification

Other than that the research meets all applicable standards for the ethics of experimentation and research integrity and seems to adhere to appropriate reporting guidelines and community standards for data availability.

Reviewer #2: Chun and colleagues present a single-center retrospective study from Korea of the ability of a variety of general and kidney-specific disease severity scores to predict mortality in patients with acute kidney injury (AKI) requiring continuous renal replacement therapy (CRRT). In short, though they somewhat overstate the utility of the kidney-specific scores, they found no single score performs that well (i.e., none had excellent discrimination nor calibration), though overall the kidney-specific scores performed somewhat better. Though the study has a lot of limitations, it is a relevant topic and reasonably thought provoking. The methods used appear appropriate and straightforward, and the data are presented clearly. However, the manuscript needs significant editing to improve the quality and clarify of the English language. In addition, I have the following recommendations, questions, and concerns:

1. The authors overstate how good any of these models are. For example, in the results the authors state that, "The AUROC revealed high discrimination ability of Demirjian’s score followed by Liano's score." In the discussion, they state that, "The AUROC for in-hospital mortality revealed high discrimination ability of Demirjian’s score (0.770) followed by Liano's score (0.728) and APACHE II score (0.710)." I would not agree with these statements. Though it's somewhat arbitrary, most would consider AUROC of 0.7-0.8 to have "moderate" or "acceptable" predictive ability. For example, a frequently cited interpretation of AUROC is as follows:

0.5 = No discrimination

0.5-0.7 = Poor discrimination

0.7-0.8 = Acceptable discrimination

0.8-0.9= Excellent discrimination

>0.9 = Outstanding discrimination

[From: Hosmer, D.W., Jr., Lemeshow, S. and Sturdivant, R.X. (2013). Assessing the Fit of the Model. In Applied Logistic Regression (eds D.W. Hosmer, S. Lemeshow and R.X. Sturdivant). https://doi.org/10.1002/9781118548387.ch5]

Overall, the results of this study are disappointing -- only 1 test was had borderline good AUC (Demirjian’s), but even that test had poor fit. I think the authors needs to do a better job of explaining that none of these tests are particularly good, and therefore they should more strongly emphasize that the primary conclusion of this study is that additional research is needed to develop better scoring systems.

2. Why was the mortality of the cohort so high? The authors should try to better address this. The authors suggest that AKI treated specifically with CRRT has higher mortality than AKI patients overall, which is true, but even for AKI requiring CRRT this is high. In most studies, the mortality of AKI requiring CRRT is closer to 50%. Comparing it to the BEST study (reference 6) -- which reported data from patients treated >20 years -- doesn't seem to adequate address the extremely high mortality seen here. To the degree that these authors are looking to evaluate mortality prediction tools, analyzing a cohort that has a much higher mortality than other centers could make the findings less generalizable. To help address this, they should report the mean or median and distribution of the disease severity scores of this cohort, particularly those that are a reflection of overall disease severity (i.e., SOFA, SAPS 3, APACHE 2, MODS). Regardless of whether they are able to somehow justify such a high mortality, this high mortality should be included as a potential limitation of the study, because these results may not be as relevant to centers with different patient selection and/or better CRRT outcomes.

3. This is extremely poorly worded: "Except for Liano's score (χ2=7.555, 166 p=0.478) and PICARD score (χ2=14.835, p=0.062), the Hosmer-Lemeshow test for in-hospital mortality demonstrated that calibration ability of all scores was not significant." Essentially, the authors are misusing the word significant here. The term "significant" when describing a statistical test indicates you reject the null hypothesis because the observed findings are unlikely to be due to change (i.e., p is <0.05). Technically, for Liano and PICARD in this study, the results of the Hosmer-Lemeshow test are *not* statistically significant (i.e., the null hypothesis that the predicted and observed outcomes are the same is not rejected). For Hosmer-Lemeshow test, a significant test indicates that the model is *not* a good fit, and a non-significant test indicates a good fit. I would suggest the authors report the results of the Hosmer-Lemeshow tests by simply providing the appropriate interpretation and avoiding the terms "significant" or "not significant", which are confusing in general for this type of test. For example, the authors could state, "Except for Liano's score (χ2=7.555, 166 p=0.478) and PICARD score (χ2=14.835, p=0.062), the Hosmer-Lemeshow test for in-hospital mortality demonstrated poor fit of the prediction models (p <0.05)." Similarly, this statement, "The APACHE II score was more discriminative compared to other general scores, but the results of the fitness test were not significant" could be changed to "The APACHE II score was more discriminative compared to other general scores, but the results of the fitness test showed poor calibration." Similarly, the abstract should state, "The Hosmer-Lemeshow test demonstrated good fit of Liano's score and PICARD scores." Likely, in the discussion I suggest, "The Hosmer-Lemeshow test on Demirjian’s score showed poor fit in our analysis, but it was more acceptable compared to the general severity scores."

4. Why are the authors publishing retrospective data from 2016-2018 in 2023? That seems odd. It certainly takes time to do this research, but a 4-year interval seems much. Is this a secondary analysis of a cohort which was generated for a prior study? If so, the authors should cite the prior work. Otherwise, at least a brief explanation as to why these patients from a cohort that is >4y old at publication seems warranted.

5. Table 3, as written, is somewhat confusing. It is hard to follow the directionality of comparisons. To help make it easier to follow, I suggest putting the score with the highest AUC first (Demirjian), followed by the second highest (Liano), etc. That should make it a little easier to read.

6. In the introduction, I found this statement to be somewhat misleading: "This is because the therapy is hemodynamically more stable than the intermittent hemodialysis therapy, and it is easy to control fluid balance and to correct metabolic acidosis or electrolyte imbalance and to correct nutritional deficiency [9]." In general, intermittent HD corrects acidosis and electrolytes just as well as CRRT. I also don't understand what is meant by "correct nutritional deficiency". I would simply end the sentence after "...control fluid balance." If they want to claim that CRRT is better for acidosis or electrolytes, a much more complicated discussion about instantaneous clearance vs. today daily dose of RRT (i.e., equilibrated Kt/V) would be needed, but it’s just best to avoid suggesting that CRRT is better than IHD for acidosis or electrolytes.

7. For the calibration tests, why did the author present results for only 5 of the 7 scoring systems? They should present them all or explain why they excluded SOFA and MODS.

8. In the discussion, for all the other prediction scores used (e.g., Mehta, Chertow, Paganini, SHARF II, and VELLORE), the authors should cite the original publications describing these scoring systems.

9. Do the authors have the distribution of baseline creatinine or eGFR? Or do they have the proportion of patients with advanced (e.g., stage 4 or 5) CKD? The authors suggest some patients with advanced CKD were included in the cohort. These patients (which many of which may be better classified as new ESKD rather than AKI) could be vastly different than patients with AKI. To better understand this cohort, more information about the relative number of patients with de novo AKI, AKI on CKD, and progressive CKD would be good.

6. PLOS authors have the option to publish the peer review history of their article (what does this mean?). If published, this will include your full peer review and any attached files.

Reviewer #1: **Yes: **Anirban Ganguli

Reviewer #2: **Yes: **J. Pedro Teixeira

---

## [Author Response · Author response to Decision Letter 0]

27 Mar 2023

Response to reviewer(s)’ comments

Dear Reviewer(s),

We are grateful for the valuable comments. After thorough discussions, we have revised our paper to reflect your helpful recommendations. Our point-by-point responses to the comments appear in blue in the following, and the revised manuscript can be found enclosed with the submission. 

Reviewer 1

The study by Chun et al is an original research to attempt using well validated scoring systems to predict 7 day and in hospital mortality in patients on CRRT. The study does follow the necessary statistical requirement to compare scoring systems and uses the standard metrics to judge prognostic scoring systems including discrimination and calibration. Unfortunately the conclusions are not valid since there are methodological flaws. Firstly many of the scores being compared have used in their developmental cohort patients only with AKI needing CRRT unlike authors study where CKD patients needing dialysis is also included. Additionally, they do not mention what percentage of their patients had CKD among survivors vs non-survivors.

Thank you for your thorough comments. Many studies included all patients diagnosed with AKI who received renal replacement therapy such intermittent hemodialysis. Since our study focused on CRRT, we included CKD patients who received CRRT, except for those who received renal replacement therapy such as intermittent hemodialysis, peritoneal dialysis, or kidney transplants. As you mentioned, we added proportion of CKD patients as following:

(Line 123-125) CKD patients accounted for 11.9% of the total study population, and there was no significant difference in the proportion of survivors and non-survivors. 

(Table 1: Line 130-135)

 Table 1. Demographics and Clinical Characteristics of Study Population

Besides this there is no mention of timing of scoring with respect to CRRT start.

Following your recommendation, we added the timing of scoring as following: 

(Line 105-107) Severity scoring was performed when it was decided to start CRRT, and all variables for severity scores were collected within 24 hours prior to the initiation of CRRT.

Minor issues: Sentences need to structured for more clarity: for instance

Line 71 : which study is mentioned here

Line 72: Another several studies is grammatically incorrect. I think authors want to say that other studies have compared predictive power of disease severity scoring systems

Line 74 : needs clarification

We appreciate for your help. We rephrased the sentences as the following: 

(Line 68-70) In addition, in studies to improve the quality of CRRT, such as the optimal start time of CRRT [25, 26],

(Line 71-72) Other studies have compared predictive power of severity scoring systems in patients with AKI. 

(Line 72-74) However, many studies included all patients diagnosed with AKI who received renal replacement therapy such intermittent hemodialysis, and few studies focused solely on CRRT [28-34]. 

Reviewer 2

1. The authors overstate how good any of these models are. For example, in the results the authors state that, "The AUROC revealed high discrimination ability of Demirjian’s score followed by Liano's score." In the discussion, they state that, "The AUROC for in-hospital mortality revealed high discrimination ability of Demirjian’s score (0.770) followed by Liano's score (0.728) and APACHE II score (0.710)." I would not agree with these statements. Though it's somewhat arbitrary, most would consider AUROC of 0.7-0.8 to have "moderate" or "acceptable" predictive ability. For example, a frequently cited interpretation of AUROC is as follows:

0.5 = No discrimination

0.5-0.7 = Poor discrimination

0.7-0.8 = Acceptable discrimination

0.8-0.9= Excellent discrimination

>0.9 = Outstanding discrimination

[From: Hosmer, D.W., Jr., Lemeshow, S. and Sturdivant, R.X. (2013). Assessing the Fit of the Model. In Applied Logistic Regression (eds D.W. Hosmer, S. Lemeshow and R.X. Sturdivant). https://doi.org/10.1002/9781118548387.ch5]

Overall, the results of this study are disappointing -- only 1 test was had borderline good AUC (Demirjian’s), but even that test had poor fit. I think the authors needs to do a better job of explaining that none of these tests are particularly good, and therefore they should more strongly emphasize that the primary conclusion of this study is that additional research is needed to develop better scoring systems.

Thank you for your careful review of our work and your suggestions regarding the manuscript. Following your recommendation, we rephrased the sentences and added the opinion in the conclusion as the following: 

(Line 159-160) The AUROC revealed acceptable discrimination ability of Demirjian’s score followed by Liano's score.

(Line 179-180) Same as in-hospital mortality, Demirjian’s score showed relatively high value to predict the 7-day mortality. 

(Line 197-198) The AUROC for in-hospital mortality revealed acceptable discrimination ability of Demirjian’s score

(Line 281-285) However, none of those evaluated in this study showed both excellent differentiation and suitability. In conclusion, all severity scoring systems included in this study were inappropriate for predicting mortality of critically ill patients requiring CRRT. Therefore, we emphasize the need to develop a novel severity scores with good calibration and high discrimination for the patients. 

2. Why was the mortality of the cohort so high? The authors should try to better address this. The authors suggest that AKI treated specifically with CRRT has higher mortality than AKI patients overall, which is true, but even for AKI requiring CRRT this is high. In most studies, the mortality of AKI requiring CRRT is closer to 50%. Comparing it to the BEST study (reference 6) -- which reported data from patients treated >20 years -- doesn't seem to adequate address the extremely high mortality seen here. To the degree that these authors are looking to evaluate mortality prediction tools, analyzing a cohort that has a much higher mortality than other centers could make the findings less generalizable. To help address this, they should report the mean or median and distribution of the disease severity scores of this cohort, particularly those that are a reflection of overall disease severity (i.e., SOFA, SAPS 3, APACHE 2, MODS). Regardless of whether they are able to somehow justify such a high mortality, this high mortality should be included as a potential limitation of the study, because these results may not be as relevant to centers with different patient selection and/or better CRRT outcomes.

Thank you for your help. We added the mean severity scores of the study population. Also we added that high mortality is a potential limitation of our study as the following: 

(Line 145-148) The mean severity scores of the study population were APACHE II score 35.5, SAPS 3 84.6, SOFA score 9.0, MODS 10.7, Liano's score 0.55, PICARD score 0.43, and Demirjian's score 0.60. Table 3 shows the comparison of severity score between survivors and non-survivors. There was significant difference in the mean of all severity scores between survivors and non-survivors (all p <0.01).

(Table 3: Line 150-155)

Table 3. Comparison of severity score between survivors and non-survivors

3. This is extremely poorly worded: "Except for Liano's score (χ2=7.555, 166 p=0.478) and PICARD score (χ2=14.835, p=0.062), the Hosmer-Lemeshow test for in-hospital mortality demonstrated that calibration ability of all scores was not significant." Essentially, the authors are misusing the word significant here. The term "significant" when describing a statistical test indicates you reject the null hypothesis because the observed findings are unlikely to be due to change (i.e., p is <0.05). Technically, for Liano and PICARD in this study, the results of the Hosmer-Lemeshow test are *not* statistically significant (i.e., the null hypothesis that the predicted and observed outcomes are the same is not rejected). For Hosmer-Lemeshow test, a significant test indicates that the model is *not* a good fit, and a non-significant test indicates a good fit. I would suggest the authors report the results of the Hosmer-Lemeshow tests by simply providing the appropriate interpretation and avoiding the terms "significant" or "not significant", which are confusing in general for this type of test. For example, the authors could state, "Except for Liano's score (χ2=7.555, 166 p=0.478) and PICARD score (χ2=14.835, p=0.062), the Hosmer-Lemeshow test for in-hospital mortality demonstrated poor fit of the prediction models (p <0.05)." Similarly, this statement, "The APACHE II score was more discriminative compared to other general scores, but the results of the fitness test were not significant" could be changed to "The APACHE II score was more discriminative compared to other general scores, but the results of the fitness test showed poor calibration." Similarly, the abstract should state, "The Hosmer-Lemeshow test demonstrated good fit of Liano's score and PICARD scores." Likely, in the discussion I suggest, "The Hosmer-Lemeshow test on Demirjian’s score showed poor fit in our analysis, but it was more acceptable compared to the general severity scores."

We appreciate for your help. We rephrased the sentences as the following: 

(Line 185-186) Except for Liano's score (χ2=7.555, p=0.478) and PICARD score (χ2=14.835, p=0.062), the Hosmer-Lemeshow test for in-hospital mortality demonstrated poor fit of the prediction models (p <0.05).

(Line 202-204) In addition, the Hosmer-Lemeshow test results of five scores, which provide predicted mortality, showed poor calibration of all scores except Liano's and PICARD scores.

(Line 216-217) The APACHE II score was more discriminative compared to other general scores, but the results of the fitness test showed poor calibration, 

(Line 40-42) The Hosmer-Lemeshow test on Demirjian’s score showed poor fit in our analysis, but it was more acceptable compared to the general severity scores.

4. Why are the authors publishing retrospective data from 2016-2018 in 2023? That seems odd. It certainly takes time to do this research, but a 4-year interval seems much. Is this a secondary analysis of a cohort which was generated for a prior study? If so, the authors should cite the prior work. Otherwise, at least a brief explanation as to why these patients from a cohort that is >4y old at publication seems warranted.

We agree with your opinion. The reason we used the data from 2016 to 2018 is that the data at that time was accessible. Since the author belonged to the department at the time, it was easy to access the data at the time. After data collection, an actual analysis was conducted from 2019 to 2020. Above all, the authors believe that the characteristics of the cohort in this study, which evaluates the adequacy of scoring systems for a particular group, will not change the relevant risk factors of the study results.

5. Table 3, as written, is somewhat confusing. It is hard to follow the directionality of comparisons. To help make it easier to follow, I suggest putting the score with the highest AUC first (Demirjian), followed by the second highest (Liano), etc. That should make it a little easier to read.

Thank you for your thorough comments. As you mentioned, we rephrased the table as the following: 

(Table 4: Line 167-172) 

Table 4. Pairwise Comparison of Receiver Operating Characteristic Curves for In-hospital Mortality

6. In the introduction, I found this statement to be somewhat misleading: "This is because the therapy is hemodynamically more stable than the intermittent hemodialysis therapy, and it is easy to control fluid balance and to correct metabolic acidosis or electrolyte imbalance and to correct nutritional deficiency [9]." In general, intermittent HD corrects acidosis and electrolytes just as well as CRRT. I also don't understand what is meant by "correct nutritional deficiency". I would simply end the sentence after "...control fluid balance." If they want to claim that CRRT is better for acidosis or electrolytes, a much more complicated discussion about instantaneous clearance vs. today daily dose of RRT (i.e., equilibrated Kt/V) would be needed, but it’s just best to avoid suggesting that CRRT is better than IHD for acidosis or electrolytes.

We appreciate your suggestion. We rephrased the sentence as the following: 

(Line 54-55) This is because the therapy is hemodynamically more stable than the intermittent hemodialysis therapy, and it is easy to control fluid balance [9].

7. For the calibration tests, why did the author present results for only 5 of the 7 scoring systems? They should present them all or explain why they excluded SOFA and MODS.

We added the reasons for exclusion as following: 

(Line 183-184) Two severity scoring systems were excluded, SOFA score and MODS, which do not generate probability of death, only counting points.

8. In the discussion, for all the other prediction scores used (e.g., Mehta, Chertow, Paganini, SHARF II, and VELLORE), the authors should cite the original publications describing these scoring systems.

Additional references is added as following: 

(Line 231-232) Mehta score [35], Liano's score, Chertow score [36]

(Line 236) Logistic Organ Dysfunction [38], Organ Dysfunction and Infection [39]

(Line 242) SHARF II score [40], PICARD score, VELLORE score [41],

(Line 372-398)

35. Mehta RL, Pascual MT, Gruta CG, Zhuang S, Chertow GM. Refining predictive models in critically ill patients with acute renal failure. J Am Soc Nephrol. 2002; 13:1350–7.

36. Chertow GM, Lazarus JM, Paganini EP, Allgren RL, Lafayette RA, Sayegh MH. Predictors of mortality and the provision of dialysis in patients with acute tubular necrosis. The Auriculin Anaritide Acute Renal Failure Study Group. J Am Soc Nephrol. 1998;9:692–8.

37. Paganini EP, Halstenberg WK, Goormastic M. Risk modeling in acute renal failure requiring dialysis: The introduction of a new model. Clin Nephrol. 1996; 46:206–11.

38. Le Gall JR, Klar J, Lemesho S, Saulnier F, Alberti C, Artigas A, et al. The Logistic Organ Dysfunction System. A New Way to Assess Organ Dysfunction in the Intensive Care Unit. JAMA. 1996;276(10):802-10.

39. Fagon JY, Chastre J, Novara A, Medioni P, Gibert C. Characterization of intensive care unit patients using a model based on the presence or absence of organ dysfunctions and/or infection: The ODIN model. Intensive Care Medicine. 1993;19:137-44.

40. Lins RL, Elseviers MM, Daelemans R, Arnouts P, Billiouw JM, Couttenye M, et al. Re-evaluation and modification of the stuivenberg hospital acute renal failure (SHARF) scoring system for the prognosis of acute renal failure: an independent multicentre, prospective study. Nephrol Dial Transplant. 2004;19:2282–8.

41. Dharan KS, John GT, Antonisamy B, Kirubakaran MG, Jacob CK. Prediction of mortality in acute renal failure in the tropics. Ren Fail. 2005;27:289–96. 

9. Do the authors have the distribution of baseline creatinine or eGFR? Or do they have the proportion of patients with advanced (e.g., stage 4 or 5) CKD? The authors suggest some patients with advanced CKD were included in the cohort. These patients (which many of which may be better classified as new ESKD rather than AKI) could be vastly different than patients with AKI. To better understand this cohort, more information about the relative number of patients with de novo AKI, AKI on CKD, and progressive CKD would be good.

Thank you for your careful comments. We provided the proportion of CKD patients regarding your comments. We also added this characteristic of the cohort as a limitation of our study as following:

(Line 123-125) CKD patients accounted for 11.9% of the total study population, and there was no significant difference in the proportion of survivors and non-survivors. 

(Table 1: Line 130-135)

Table 1. Demographics and Clinical Characteristics of Study Population

(Line 273-277) In this study, CKD patients were included in the study population, except for patients who received renal replacement therapy such as intermittent hemodialysis, peritoneal dialysis, and kidney transplantation. Since the focus is on patients receiving CRRT, the study results are unlikely to change due to the characteristics of this cohort, but the lack of information such as baseline creatinine or eGFR is a limitation of this study.

---

## [Decision Letter · Decision Letter 1]

14 Apr 2023

PONE-D-22-26854R1Assessment of severity scoring systems for predicting mortality in critically ill patients receiving continuous renal replacement therapyPLOS ONE

Dear Dr. Chun,

Thank you for submitting your manuscript to PLOS ONE. After careful consideration, we feel that it has merit but does not fully meet PLOS ONE’s publication criteria as it currently stands. Therefore, we invite you to submit a revised version of the manuscript that addresses the points raised during the review process.

**The revised manuscript is significantly improved. Most of the reviewer’s comments have been addressed. However, the authors need to further consider the few remaining minor recommendations by Reviewer 2 dealing with the conclusions. Moreover, additional English language editing of the manuscript is required.**

We look forward to receiving your revised manuscript.

Kind regards,

Giuseppe Remuzzi

Academic Editor

PLOS ONE

Journal Requirements:

Reviewers' comments:

Reviewer's Responses to Questions

**Comments to the Author**

1. If the authors have adequately addressed your comments raised in a previous round of review and you feel that this manuscript is now acceptable for publication, you may indicate that here to bypass the “Comments to the Author” section, enter your conflict of interest statement in the “Confidential to Editor” section, and submit your "Accept" recommendation.

Reviewer #1: All comments have been addressed

Reviewer #2: (No Response)

2. Is the manuscript technically sound, and do the data support the conclusions?

Reviewer #1: Yes

Reviewer #2: Yes

3. Has the statistical analysis been performed appropriately and rigorously? 

Reviewer #1: Yes

Reviewer #2: Yes

4. Have the authors made all data underlying the findings in their manuscript fully available?

Reviewer #1: Yes

Reviewer #2: Yes

5. Is the manuscript presented in an intelligible fashion and written in standard English?

Reviewer #1: Yes

Reviewer #2: Yes

6. Review Comments to the Author

Reviewer #1: (No Response)

Reviewer #2: I think the manuscript is significantly improved. I have few additional minor recommendations:

1. In the discussion [lines 263-265 in the clean version] I once again suggest , "The Hosmer-Lemeshow test on Demirjian’s score showed poor fit in our analysis, but it was more acceptable compared to the general severity scores."

2. The term suitability is vague. In the conclusions, I recommend changing, "However, none of those evaluated in this study showed both excellent differentiation and suitability" to a more precise statement, "However, none of those evaluated in this study showed both excellent differentiation and calibration."

3. In the conclusion I also suggest being a bit less extreme by stating, "In conclusion, all severity scoring systems included in this study had limited ability to predict mortality of critically ill patients requiring CRRT."

4. As I commented in my first review, I think the manuscript could significantly benefit from additional English language editing.

7. PLOS authors have the option to publish the peer review history of their article (what does this mean?). If published, this will include your full peer review and any attached files.

Reviewer #1: **Yes: **Anirban Ganguli

Reviewer #2: **Yes: **J. Pedro Teixeira

---

## [Author Response · Author response to Decision Letter 1]

24 Apr 2023

Dear Reviewers (s):

We are grateful for your insightful comments. After thorough discussions, we have revised our paper to reflect your valuable recommendations. Our point-by-point responses to the comments are provided in blue in the following text, and the revised manuscript is enclosed with the submission.

Reviewer 2

1. In the discussion [lines 263-265 in the clean version] I once again suggest, "The Hosmer-Lemeshow test on Demirjian’s score showed poor fit in our analysis, but it was more acceptable compared to the general severity scores."

Response: Thank you for your careful review of our work and for your suggestions regarding this manuscript. Following your recommendation, we have rephrased the sentence as follows: 

(Line 270-271) The Hosmer-Lemeshow test on Demirjian’s score showed a poor fit in our analysis; however, it was more acceptable than general severity scores. 

2. The term suitability is vague. In the conclusions, I recommend changing, "However, none of those evaluated in this study showed both excellent differentiation and suitability" to a more precise statement, "However, none of those evaluated in this study showed both excellent differentiation and calibration."

Response: Thank you for your insightful comment. We have rephrased the sentence as follows: 

(Line 292-293) However, none of the parameters evaluated in this study exhibited both excellent differentiation and calibration.

3. In the conclusion I also suggest being a bit less extreme by stating, "In conclusion, all severity scoring systems included in this study had limited ability to predict mortality of critically ill patients requiring CRRT." 

Response: Thank you for your valuable comments. As per your suggestion, we have rephrased the sentence as follows: 

(Line 294-295) In conclusion, all severity scoring systems included in this study had a limited ability to predict mortality in critically ill patients requiring CRRT.

4. As I commented in my first review, I think the manuscript could significantly benefit from additional English language editing.

Response: Thank you for the valuable suggestion. We would like to thank Editage (www.editage.co.kr) for reviewing and editing the revised manuscript for English language.

---

## [Decision Letter · Decision Letter 2]

12 May 2023

Assessment of severity scoring systems for predicting mortality in critically ill patients receiving continuous renal replacement therapy

PONE-D-22-26854R2

Dear Dr. Chun,

We’re pleased to inform you that your manuscript has been judged scientifically suitable for publication and will be formally accepted for publication once it meets all outstanding technical requirements.

**The re-revised version of the manuscript is definitely improved. The authors have properly addressed all the remaining issues raised by the Reviewer 2, and previously all comments from Reviewer 1. Thus, the manuscript is now suitable for publication.**

Kind regards,

Giuseppe Remuzzi

Academic Editor

PLOS ONE

Additional Editor Comments (optional):

Reviewers' comments:

Reviewer's Responses to Questions

**Comments to the Author**

1. If the authors have adequately addressed your comments raised in a previous round of review and you feel that this manuscript is now acceptable for publication, you may indicate that here to bypass the “Comments to the Author” section, enter your conflict of interest statement in the “Confidential to Editor” section, and submit your "Accept" recommendation.

Reviewer #2: All comments have been addressed

2. Is the manuscript technically sound, and do the data support the conclusions?

Reviewer #2: Yes

3. Has the statistical analysis been performed appropriately and rigorously? 

Reviewer #2: Yes

4. Have the authors made all data underlying the findings in their manuscript fully available?

Reviewer #2: Yes

5. Is the manuscript presented in an intelligible fashion and written in standard English?

Reviewer #2: Yes

6. Review Comments to the Author

Reviewer #2: The manuscript is significantly improved with the changes made. I have no additional comments to make at this point.

7. PLOS authors have the option to publish the peer review history of their article (what does this mean?). If published, this will include your full peer review and any attached files.

Reviewer #2: **Yes: **J. Pedro Teixeira

---

## [Editor Report · Acceptance letter]

17 May 2023

PONE-D-22-26854R2 

Assessment of severity scoring systems for predicting mortality in critically ill patients receiving continuous renal replacement therapy 

Dear Dr. Chun:

I'm pleased to inform you that your manuscript has been deemed suitable for publication in PLOS ONE. Congratulations! Your manuscript is now with our production department. 

Kind regards, 

on behalf of

Prof. Giuseppe Remuzzi 

Academic Editor

PLOS ONE